# Typhoon damage on a shallow mesophotic reef in Okinawa, Japan

Kristine N. White[1], Taku Ohara[2,3], Takuma Fujii[2], Iori Kawamura[2], Masaru Mizuyama[2], Javier Montenegro[2], Haruka Shikiba[2], Tohru Naruse[4], TY McClelland[2], Vianney Denis[5] and James D. Reimer[2]

[1] University of Maryland University College, Asia Division, Camp Foster Education Center, United States/Japan
[2] Graduate School of Engineering and Science, University of the Ryukyus, Okinawa, Japan
[3] Benthos Divers, Okinawa, Japan
[4] Tropical Biosphere Research Center, Iriomote Station, University of the Ryukyus, Okinawa, Japan
[5] Biodiversity Research Center, Academia Sinica, Taipei, Taiwan

## ABSTRACT

Little is known about effects of large storm systems on mesophotic reefs. This study reports on how Typhoon 17 (Jelawat) affected Ryugu Reef on Okinawa-jima, Japan in September 2012. Benthic communities were surveyed before and after the typhoon using line intercept transect method. Comparison of the benthic assemblages showed highly significant differences in coral coverage at depths of 25–32 m before and after Typhoon 17. A large deep stand of *Pachyseris foliosa* was apparently less resistant to the storm than the shallower high diversity area of this reef. Contradictory to common perception, this research shows that large foliose corals at deeper depths are just as susceptible to typhoon damage as shallower branching corals. However, descriptive functional group analyses resulted in only minor changes after the disturbance, suggesting the high likelihood of recovery and the high resilience capacity of this mesophotic reef.

Corresponding author
Kristine N. White,
kristine.white@faculty.umuc.edu

## INTRODUCTION

Typhoon damage from direct physical disturbances, turbidity, sedimentation, and salinity changes can be destructive to shallow coral reefs and has been well studied (*Van Woesik, Ayling & Mapstone, 1991*; *Harmelin-Vivien, 1994*; *Ninio et al., 2000*; *Cheal et al., 2002*; *Hongo, Kawamata & Goto, 2012*). Declines in coral cover on shallow reefs (<25 m in depth) have been documented, specifically in genera such as *Acropora*, *Montastraea*, *Porites*, *Agaracia*, *Diploria*, *Millepora*, *Siderastrea*, *Pocillopora*, *Pachyseris*, *Montipora*, and *Merulina* (*Harmelin-Vivien, 1994*; *Van Woesik, De Vantier & Glazebrook, 1995*; *Fabricius et al., 2008*). Although massive corals such as *Montipora*, *Montastraea*, *Siderastrea*, and *Diploria* can be overturned during typhoons, they are often the most resistant to storms and therefore tend to dominate or increase in cover after a disturbance (*Harmelin-Vivien, 1994*; *Fabricius et al., 2008*; *Hongo, Kawamata & Goto, 2012*). Increase in cover of genera *Porites*, *Montipora*,

and *Lobophyllia*, unattached fungiids and even *Acropora* species with high regeneration efficiencies have been documented after disturbances (*Harmelin-Vivien, 1994*; *Van Woesik, De Vantier & Glazebrook, 1995*; *Fabricius et al., 2008*; *Kuo et al., 2011*).

Recent observational data of a mesophotic reef (35–40 m) near Kume-jima, Okinawa, Japan before and after a typhoon reported that newly broken *Acropora* pieces fused to new branches, producing clones, and the reef made a quick recovery after typhoon damage (*Fujita, Kimura & Atsuo, 2012*).

However, comparatively little has been published on the effects of tropical cyclones on deeper reefs (*Randall & Eldredge, 1977*; *Woodley et al., 1981*; *Walsh, 1983*; *Pfeffer & Tribble, 1985*; *Harmelin-Vivien & Laboute, 1986*; *Van Woesik, Ayling & Mapstone, 1991*). In general, reefs at depths greater than 25 m appear to be less affected by tropical cyclones than shallower reefs (*Harmelin-Vivien, 1994*; *Bongaerts et al., 2011*; *Bridge & Guinotte, 2012*). *Harmelin-Vivien (1994)* reported that most physical damage to deep reefs was due to rolling colonies dislodged from shallower areas. However, as reported in *Harmelin-Vivien (1994)*, tropical cyclone induced coral destruction was observed to depths of 25 m in Belize (*Highsmith, Riggs & D'Antonio, 1980*), to 30 m on the Great Barrier Reef (*Van Woesik, Ayling & Mapstone, 1991*), to 30 m in Guam and Hawaii (*Walsh, 1983*), to 50 m in Jamaica (*Woodley et al., 1981*), to 50 m in Hawaii (*Pfeffer & Tribble, 1985*), to 50–65 m on the Great Barrier Reef (*Bongaerts et al., 2013*), and to 90–100 m in French Polynesia (*Laboute, 1985*; *Harmelin-Vivien & Laboute, 1986*). *Bridge & Guinotte (2012)* concluded that although depth does have an impact on coral community survival, rugosity and angle of slope play large roles in the protection of species on a reef. Large communities of broadcast spawning species are important to the recovery of reefs and are more likely to survive in deeper waters than in shallower waters during a typhoon (*Madin & Connolly, 2006*; *Bridge & Guinotte, 2012*), and recruitment of coral larvae is likely the most effective method of recovery for disturbed reefs (*Harmelin-Vivien, 1994*).

Typhoon 17 (Jelawat) struck the west coast of Okinawa-jima Island on 29 September 2012 (Fig. 1), heading from the southwest to the northeast with a general wind direction of northwest. The Japan Meteorological Agency (JMA) and the United States Navy Joint Typhoon Warning Center (JTW) documented this record-breaking typhoon as the third strongest typhoon to hit Okinawa-jima Island since weather radar observations were started in 1954 with maximum wave heights of 12 m. The Okinawa Meteorological Observatory (OMO) recorded the following maximum measurements for Typhoon 17 at the northern end of Okinawa-jima Island (Nago Meteorological Station): 32.2 m/s sustained winds, 57.4 m/s wind gusts, 947.4 hPa atmospheric pressure.

Recently, *Ohara et al. (2013)* reported on a previously undiscovered shallow mesophotic coral reef in Okinawa, Japan. The Japanese name for this reef is "Ryugu," based on its resemblance to the undersea palace of Ryūjin, the dragon god of the sea. The deeper sections of Ryugu (32–42 m) were reported to be primarily composed of *Pachyseris foliosa* (*Veron, 1990*), with shallower sections (25–32 m) showing much higher diversity (*Ohara et al., 2013*). Little is known about *Pachyseris foliosa*, although the depth range of this species

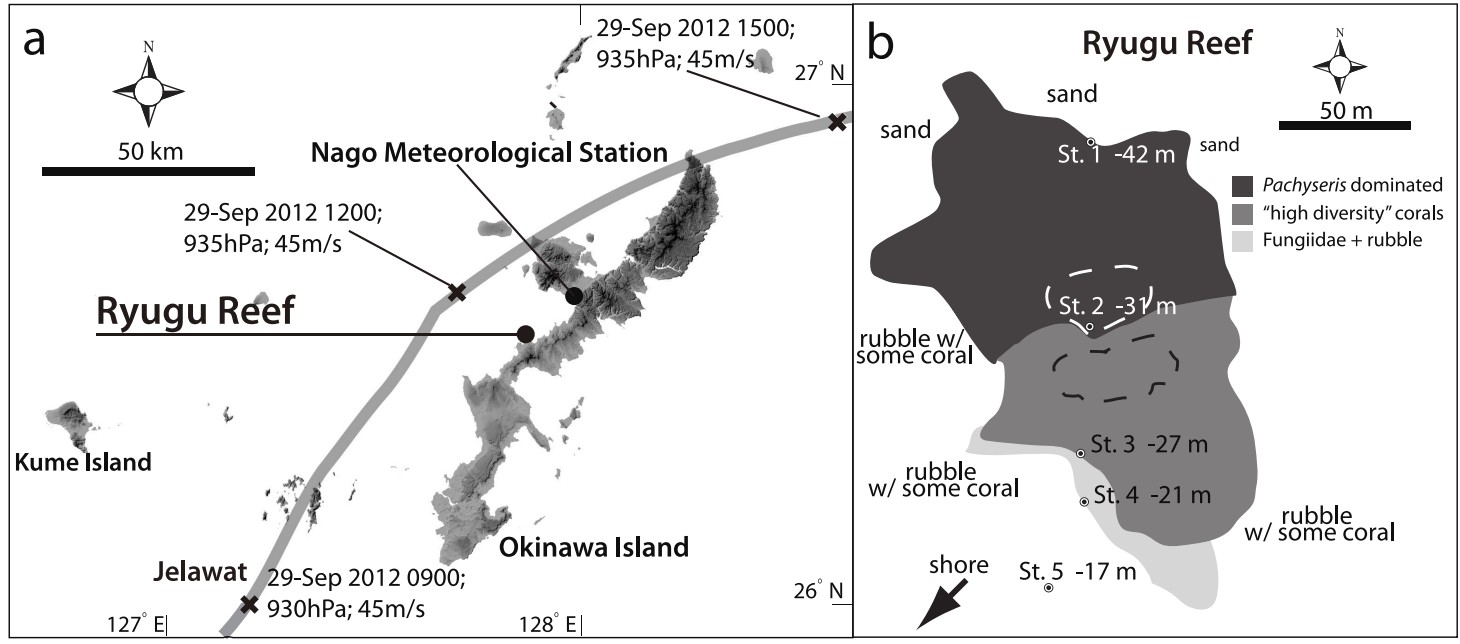

**Figure 1  Map of Typhoon track and study stations.** (A) Map showing track of Typhoon 17 (Jelawat) around Okianwa-jima Island with position and (B) studied stations of Ryugu Reef. Data on the track indicates date, time, central atmospheric pressure and maximum wind speed at each "X" mark. In (B), dotted areas indicate positions of transects.

appears to be deeper than previously reported by *Hoeksema, Rogers & Quibilan* in *2008* (25–30 m).

This study reports on how Typhoon 17 affected Ryugu Reef with before and after transect data. Objectives of this study include examination of coral communities at different depths and the identification of species and functional groups most affected by typhoons. One hypothesis tested was that shallow mesophotic reefs with large monospecific stands are more resistant to storm damage than diverse reefs.

## MATERIALS AND METHODS

Five stations were designated at the Ryugu site (Fig. 1). Station 1 was the deepest (42 m) and Station 5 the shallowest (17 m). Stations are shown in Fig. 1 and summarized in Table 1. Temperature was recorded every 30 min using temperature loggers (HOBO U22 Temp Pro v2 logger; Onset Corp., Massachusetts, USA) placed approximately 30–50 cm from the substrate at each station from 12 September 2012 to 10 January 2013.

Water motion was estimated at each station (except Station 1) by the dissolution of plaster balls. Plaster balls (10.5 cm diameter) were made following *Komatsu & Kawai (1992)*. The balls were set approximately 50 cm above the substrate at Stations 2–5 on 11 January 2013 and removed on 16 January 2013. Water speeds for each station were calculated following the equations provided in *Yokoyama, Inoue & Abo (2004)*.

Ten meter line transects were surveyed both before (17 April 2012 and 11–12 September 2012) and after (14 December 2012) Typhoon 17 at locations near Stations 2 (7 transects before, 9 transects after) and 3 (10 transects before, 8 transects after). Based on the amount

**Table 1  Stations at Ryugu Reef.**

| Station | Location/Description | Depth (m) |
|---|---|---|
| 1 | Outside outer edge of dense *Pachyseris foliosa* area, sandy | 42 |
| 2 | Upper edge of dense *Pachyseris foliosa* area | 31.2 |
| 3 | Upper edge of high diversity area | 26.5 |
| 4 | Upper edge of Fungiidae/rubble area | 21.3 |
| 5 | Sand, coral rubble | 17 |

of data available, only Stations 2 and 3 were included in analyses. For each line transect, a 10 meter tape measure was laid out along a constant depth contour and overlapping photographs or video was taken along the line. Photographs or videos taken along the transects were used to report the total distance occupied by each operational taxonomic unit (OTU) identified. When feasible, OTUs were identified to species level following *Hoeksema (1989)* and *Gittenberger, Reignen & Hoeksema (2011)* for Fungiidae and *Veron (2000)* and *Budd et al. (2012)* for other species.

Community data were analyzed using PRIMER 6 statistical software in order to find differences in coral communities before and after the typhoon (*Clarke & Warwick, 2001*). All percent cover data were square-root transformed prior to analysis to moderately down-weight the importance of large space occupying operational taxonomic units (OTU). Bray-Curtis similarity matrices were calculated at Stations 2 and 3. A one-way analysis of similarities test (ANOSIM) was performed to determine the difference and magnitude of difference in the assemblages before and after Typhoon 17. Non-Metric Multidimensional scaling (nMDS) was used to visualize multivariate patterns on the basis of the Bray-Curtis matrix. Bubble plot (square-root transformed cover data) was added to the plots to visualize variation in relevant OTUs. Each circle in Fig. 4 displays relative abundance of live coral species (based on square root transformed data of the species' occurrence along each transect). Finally, the percentage contributions of each benthic grouping for observed differences between locations were assessed with the SIMPER routine.

To assess potential changes in the functionality of the coral community after the typhoon coral taxa were classified into functional groups according to the shapes of the colonies following *Denis et al. (2013)*. Each OTU was assigned to one or more of eight functional groups: massive, encrusting, foliose, columnar, plate-like, bushy, arborescent, and unattached (Table S1). These were defined by each colony's growth form as described in *Veron (2000)*, *Wallace (1999)* and by visual observation. Functional composition of the coral assemblages were calculated based on the relative abundance of coral OTU and plot for Stations 2 and 3 before and after the typhoon.

## RESULTS

Temperature sensors showed that the temperature was typically 0.1–0.2 degrees lower at Station 5 (shallowest station) than at any of the other stations, although the temperature

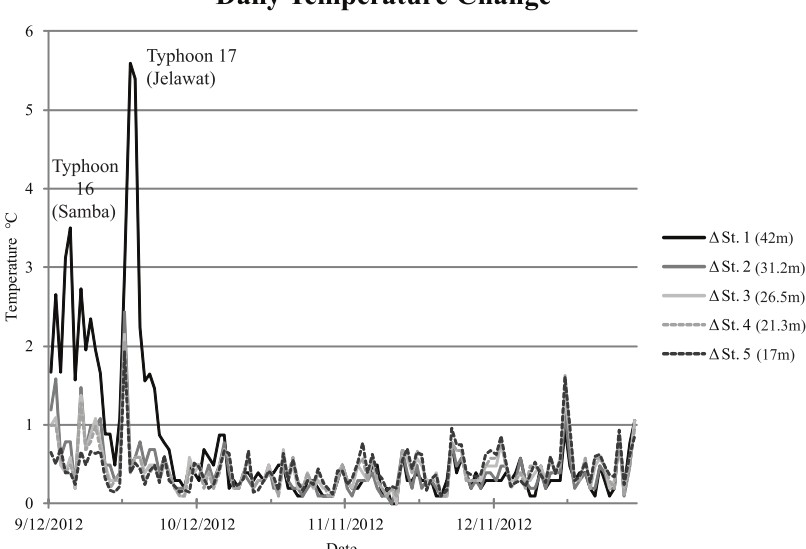

**Figure 2 Daily temperature change for Ryugu Reef.** Graph showing daily temperature change for Ryugu Reef, Stations 1–5, 12 September 2012 to 10 January 2013.

appeared to fluctuate the most at this station. Temperature drops were observed at all stations during and directly after typhoons, with the largest decrease in temperature at Station 1 (20.9°C after Typhoon 17) (Fig. 2). Based on plaster ball weight loss, Station 2 had the lowest amount of water movement compared to the other stations. The weight loss of each plaster ball and water speed for each station were as follows: Station 2: lost 254 g, 9.2 cm/s; Station 3: lost 284 g, 11.2 cm/s; Station 4: lost 292 g, 11.7 cm/s; Station 5: lost 294 g, 12.5 cm/s.

Table S1 lists all OTUs documented on transects and their percent cover change before and after Typhoon 17. Live coral cover decreased and coral rubble increased by 33.3% at Station 2 and by 11.4% at Station 3 after Typhoon 17. Figure 3 shows before and after images at Stations 2 and 3.

Composition of the benthic communities before and after Typhoon 17 (Fig. 4) presented a significant difference at both Station 2 (ANOSIM test, $R = 0.572$, $p = 0.001$) and Station 3 (ANOSIM test, $R = 0.24$, $p = 0.009$). At Station 2, the change in the occurrence of coral rubble on the transects contributed the most to this difference (Simper-test, 33.0%, Fig. 4), followed by the coverage of *Lithophyllon repanda* (12.7%), *Pachyseris foliosa* (11.6%), then *Galaxea* sp. 1 (11.2%). At Station 3, every OTU contributed to <10% of the difference observed. Change in the occurrence of coral rubble contributed to only 6% of this difference (Fig. 4). Interestingly, the difference observed at Station 3 was not significant ($R = 0.07$, $p = 0.137$) when the effects of the dominant OTU were not reduced using square root transformation. Functionality of the coral communities (Fig. 5) at both stations seems only slightly affected by the typhoon. Among the major differences observed at Station 2, the encrusting group decreased by 8%, while the foliose group increased by 15%. At Station 3, bushy (8%), columnar (6%) and plate-like (3%) groups suffered the

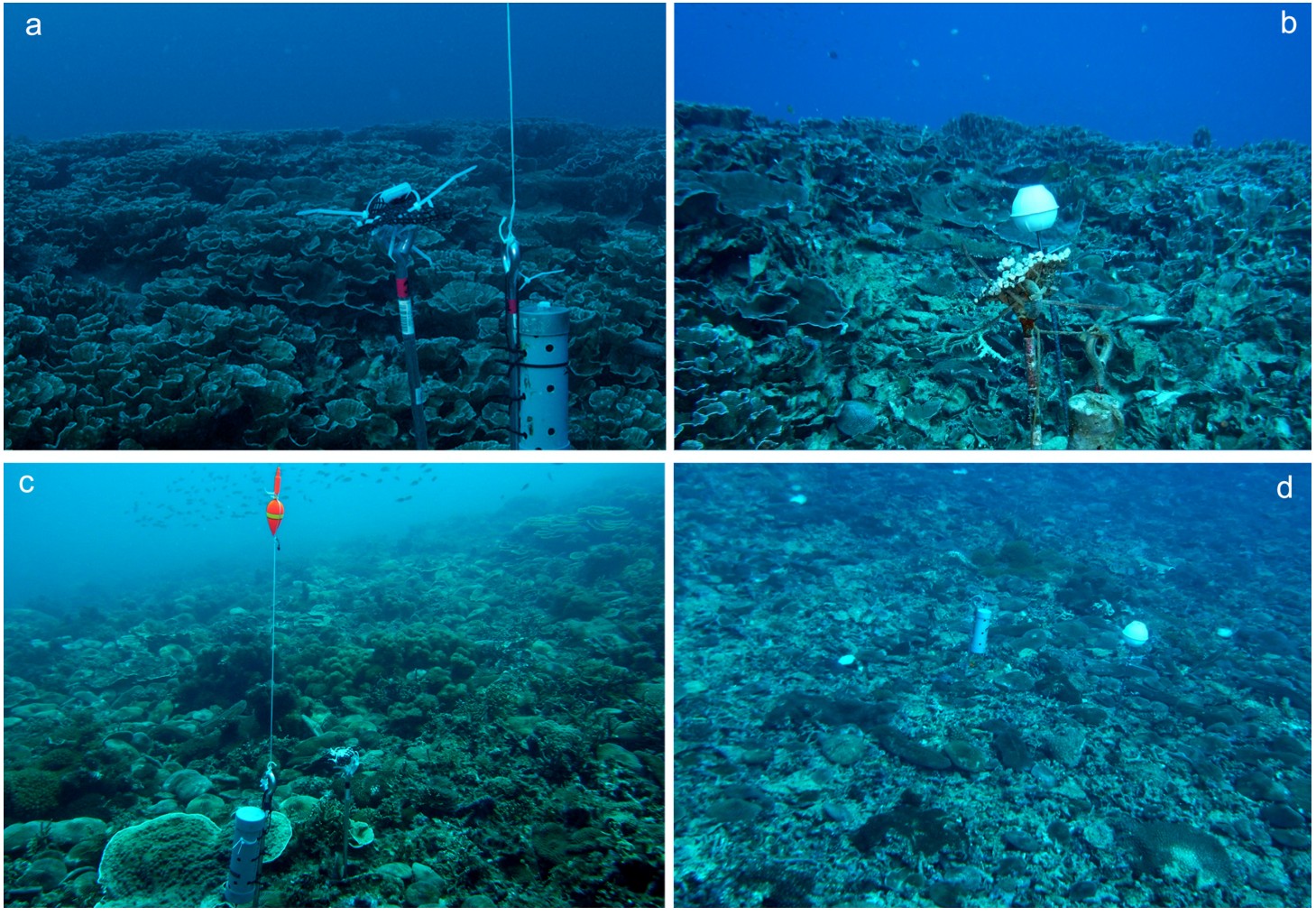

**Figure 3** Ryugu Reef photographs before and after Typhoon 17. Station 2, (A) 12 September 2012, (B) 01 January 2013; Station 3, (C) 12 September 2012, (D) 01 January 2013.

most from the path of the typhoon while encrusting (7%) and foliose (11%) corals were more resistant to this disturbance.

## DISCUSSION

Typhoon 17 resulted in highly significant changes to the live coral abundance at Ryugu Stations 2 and 3. The depth of this reef does not appear to have sheltered corals from drastic damage with notable increases in coral rubble in many of the study areas. Most interestingly, *P. foliosa* was among those species most affected by Typhoon 17. Apparently more diverse and complex communities, such as at Station 3, are more resistant to typhoons in terms of survivability and functional group distribution, perhaps due to the lower impact on individual species. Based on SIMPER tests, there were many small differences in diverse OTUs at Station 3 compared to Station 2, where only 4 OTUs contributed to 70% of the typhoon effects. Station 2, primarily composed of *P. foliosa*,

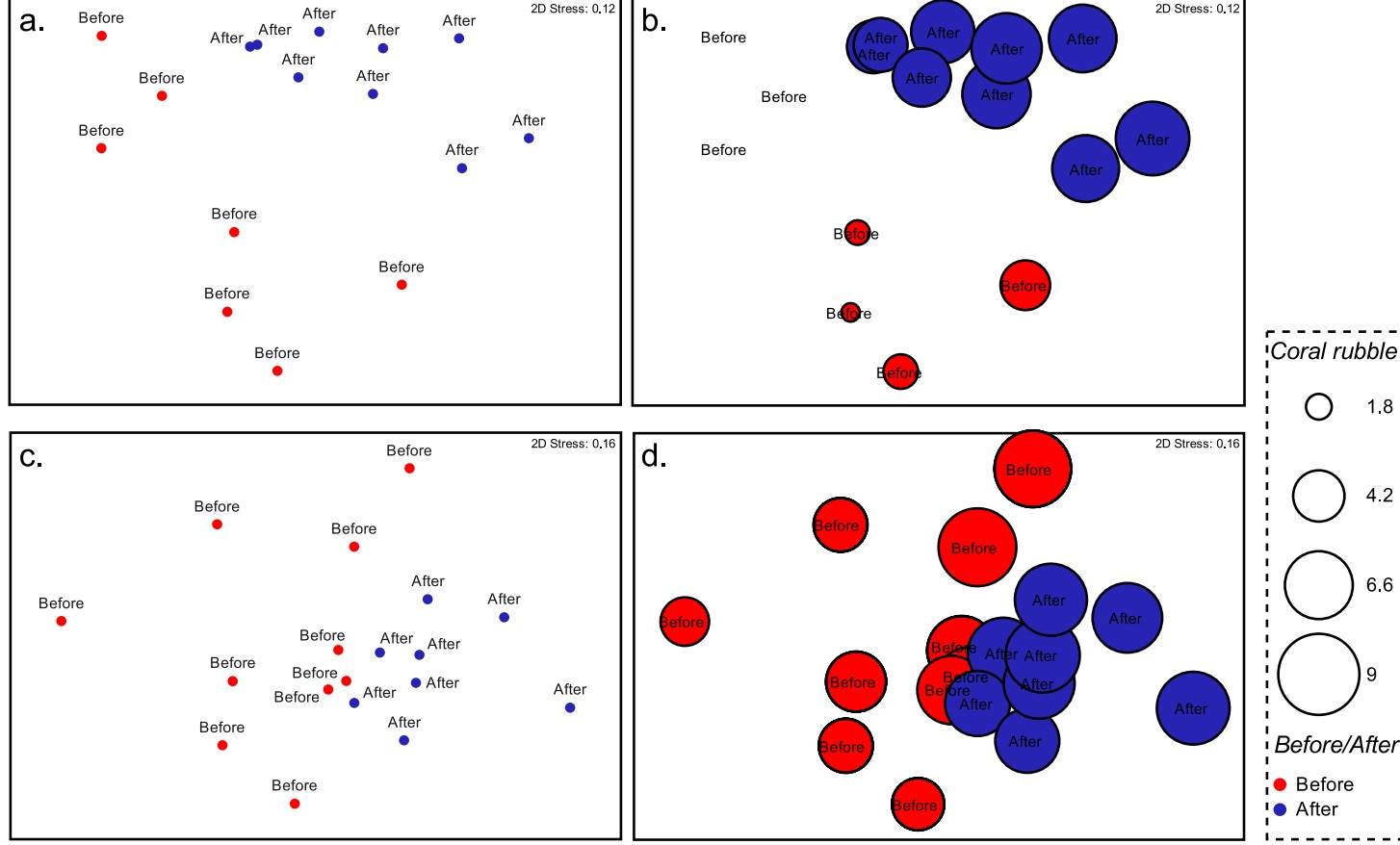

**Figure 4  Non-metric dimensional scaling of the benthic communities at Ryugu Reef.** Non-metric dimensional scaling of the benthic communities at Ryugu Reef based on the Bray-Curtis similarities matrices. Each circle in the bubble plot displays relative abundance of live coral or percent cover of coral rubble (based on square root transformed data of the species' occurrence along each transect) (A) Station 2: live coral species, (B) Station 2: coral rubble, (C) Station 3: live coral species, (D) Station 3: coral rubble. Red circle: before typhoon, blue circle: after typhoon.

was heavily impacted by this storm, despite the fact that it was deeper than Station 3, suggesting that the foliose structure of *P. foliosa* is vulnerable to physical disturbances. The large monospecific stand found here is also in a more stable environment, likely making it more sensitive to disturbance (*Hughes, 1989*; *Rogers, 1992*; *Rogers, 1993*; *Harmelin-Vivien, 1994*). Therefore, our hypothesis that shallow mesophotic reefs with large monospecific stands are more resistant to storm damage is rejected.

Consistent with previous shallow water typhoon damage studies (*Harmelin-Vivien, 1994*; *Van Woesik, De Vantier & Glazebrook, 1995*; *Fabricius et al., 2008*; *Kuo et al., 2011*) corals in the genus *Acropora* were strongly affected and were mostly dead at Ryugu after Typhoon 17, whereas unattached Fungiidae corals were mostly healthy. The fungiid corals may have been hidden under other living corals and after Typhoon 17 became more visible with the other corals having been damaged. Away from the transect locations; however, several Fungiidae corals were completely buried by newly generated *Acropora* rubble and other branching coral rubble. Accordingly, despite minor changes in the functionality of the coral community observed before and after Typhoon 17, the groups

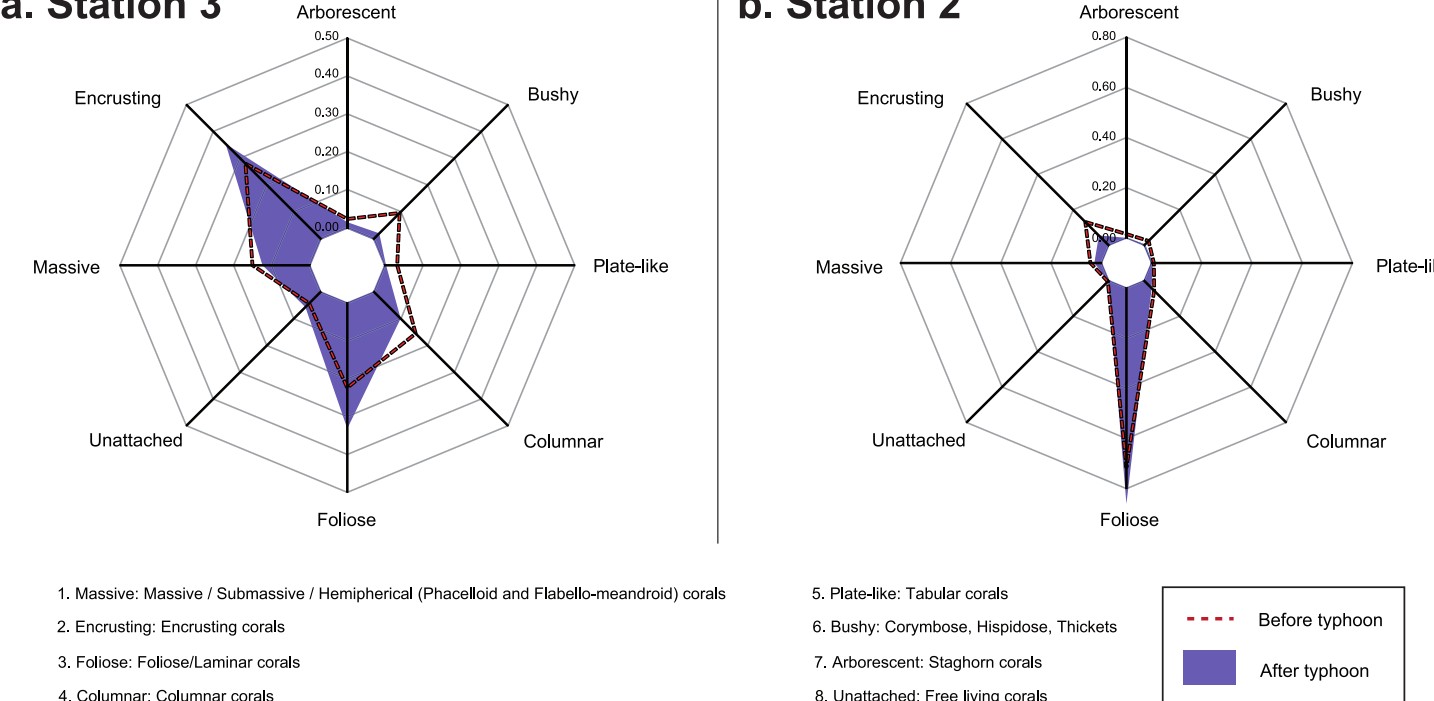

**Figure 5 Functional composition of the coral assemblage before and after Typhoon 17 at Ryugu Reef.** Functional composition of the coral assemblage before and after Typhoon 17 at Ryugu Reef, Stations 2 and 3 based on relative abundance of the coral OTUs. Axes represent the relative contribution of each of the 8 functional groups.

the most affected at Station 3 were the bushy, columnar and plate-like corals. At Station 2, the dominance of *P. foliosa* may have masked any differences. Most damaged colonies of *P. foliosa* were still alive after Typhoon 17, suggesting that species composition of this area may not change. This opposes the common idea that only massive corals would remain after disturbance (*Harmelin-Vivien, 1994*), suggesting a strong potential for the recovery and resilience of Ryugu Reef. *Pachyseris* species are typically gonochoric spawners that are most likely unable to fuse and create clones (*Richmond & Hunter, 1990*). If *Pachyseris* species reproduce only by spawning, *Pachyseris*-dominated reefs such as Ryugu should have a much slower recovery rate than *Acropora*-dominated reefs such as the one found near Kume-jima. However, recruitment of coral larvae may allow this reef to recover relatively quickly. During this study, new *P. foliosa* polyps were observed growing two to three months after Typhoon 17, suggesting that the *Pachyseris* portion of the reef had already started to recover from the damage it incurred.

Based on plaster ball data, Ryugu is a fairly calm reef and the lower currents at Station 2 may be due to less tides or wave impacts, and is worth investigating further in future studies. The lower temperature observed during Typhoon 17 at Station 1 (42 m) may be due to upwelling or thermal averaging due to wind driven vertical mixing with deeper cooler water that was enhanced by the onset of the typhoon, as seen during other large storms. Figure 2 shows large changes in temperature on 18 September (drop to 26.0°C)

and 30 September (drop to 20.9°C), both of which correspond with large typhoon systems (Fig. 1; Typhoons 16 and 17, respectively).

Many studies have found that increasing sea surface temperatures and global climate change have and will continue to cause increases in typhoon frequency, power dissipation, and storm intensity (*Emanuel, 2005*; *Trenberth, 2005*; *Webster et al., 2005*; *Emanuel, Sundararajan & Williams, 2008*; *Tu, Chou & Chu, 2009*). *Tu, Chou & Chu (2009)* have documented a northward shift in typhoon tracks in the western North Pacific-East Asia region with an increase in typhoon frequency in the Taiwan/East China Sea region (3.3 per year from 1970–1999; 5.7 per year from 2000–2006). *Emanuel (2005)* documented an increase in destructiveness of cyclones since the 1970s and has predicted a continued increase with global climate change. Global climate change is expected to bring larger and stronger typhoons to Okinawa, which will likely affect the survivability of some coral populations. A potential increase in storms makes it even more important to understand their effect on mesophotic reefs, which have been thought to act as refugia for many marine organisms during disturbances on shallow reefs. This study has shown that despite their depth, shallow mesophotic reefs may also be strongly affected by disturbances. It is, therefore, critical to document the succession of this reef after disturbances such as Typhoon 17 to understand its resilience and the role that mesophotic reefs may play in the future of coral reefs.

## ACKNOWLEDGEMENTS

Thanks go to the boat captain, Tokunobu Toyama, and diving staff, Sakiko Kawabata and Yoko Fudesaka, for their assistance during transect surveys.

### Funding

V Denis is the recipient of a Post-Doctoral fellowship by the National Science Council of Taiwan. JD Reimer was funded by the Rising Star Program, and International Research Hub Project for Climate Change and Coral Reef/Island Dynamics, both at the University of the Ryukyus. The funders had no role in study design, data collection and analysis, decision to publish, or preparation of the manuscript.

### Grant Disclosures

The following grant information was disclosed by the authors:
National Science Council of Taiwan.
Rising Star Program and International Research Hub Project for Climate Change and Coral Reef/Island Dynamics, both at the University of the Ryukyus.

### Competing Interests

There are no competing interests.

## Author Contributions

- Kristine N. White performed the experiments, analyzed the data, wrote the paper.
- Taku Ohara conceived and designed the experiments, performed the experiments, analyzed the data, contributed reagents/materials/analysis tools.
- Takuma Fujii performed the experiments, analyzed the data.
- Iori Kawamura, Masaru Mizuyama, Javier Montenegro, Haruka Shikiba, Tohru Naruse and TY McClelland performed the experiments.
- Vianney Denis analyzed the data.
- James D. Reimer conceived and designed the experiments, performed the experiments, analyzed the data, contributed reagents/materials/analysis tools, wrote the paper.

## Supplemental Information

Supplemental information for this article can be found online at http://dx.doi.org/10.7717/peerj.151.

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
