# Peer review of "Typhoon damage on a shallow mesophotic reef in Okinawa, Japan"

_PeerJ, doi:10.7717/peerj.151_

## Round 0.1 · original submission · Minor Revisions

Overall both referees were very positive about your paper and recommend acceptance. I concur with them, and am pleased to be able to tell you that your paper is acceptable for publication pending minor revisions in response to the comments raised by the referees. In particular, I concur with the comment about the quality of the figures - I am not sure if it is an issue of the PDF conversion or not, but the figures were hard to read and Figure has grey/white in a & c, but b & d appear all grey with just text "before" and "after" - some color-coding here would really help the readability of the figure.

Referee #2 suggests some additional references and asks exactly which literature to date has mischaracterized the susceptibility of deeper reefs to storm damage. I think this comment is easily dealt with by a slight revision of the text to something along the lines of it being intuitive that depth provides some buffer to storm damage or adding a reference such as Hughes & Connell (1999 L&0 44:932–940) or Scoffin (1993 Coral Reefs 12:203-221) to support your statement that there is a perception that mesophotic reefs are less susceptible to storm damage. In addition to the Harmelin-Vivien (1994) review pointed out by the referee, a more recent missing reference seems to be Bongaerts et al. 2013 Coral Reefs (Cyclone damage at mesophotic depths on Myrmidon Reef).

Finally, I also wanted to pass along the confidential comment from one suggested referee who declined to review the paper, because they are similar in tone to those of Reviewer #2. They said that it seemed to be a fine study, but they felt it was "over-sold in an attempt to ride the mesophotic bandwagon and imply that the study is more novel than it really is." They felt that the work should stand on it's own and that the novelty pitch severly detracted from the paper and they would reject it based solely on that rather than the content. I tell you this because I believe that one of the benefits of review is to get feedback about how some readers view the work, so that you have the option to revise before final publication. I am fine with whatever you decide in this regard, but wanted to pass along the information so that you can consider toning it down if you choose prior to final publication.

Aloha,
Rob

Reviewer 1 ·

Basic reporting

This study is very interesting to show that corals at deeper depth are less resistant to the storm than we expected.

Experimental design

no problem

Validity of the findings

good

Additional comments

I think that Typhoon 16 also affected the deep coral communities. I wonder why the authors consider it.

Figure 1: it is hard to see any words in this figure. Also should put "a" and "b" on the figure.

Figure 4: need much more resoluiion. Also need to explain more how to see the data. I do not understand the meaning "each circle displays relative abundance of the species" from this figure.
And what do you mean the number of coral rubble?
Also it is very complicated to see this because coral rubbles were shown by full circles, for both "before" and "after".

Minor comments
"Montastraea" is correct, not "Montastrea".
Also recently "Montastraea" was separated into three genera, Montastraea (for M. cavernosa), Phymastrea (for Indo-Pacific species), Orbicella (for M. annularis complex). If possible, consider it when the authors use "Montastraea".

·

Basic reporting

The authors present novel data measuring and analyzing coral community structure at two intermediate depths before and after a major storm/disturbance. Given the relatively depauperate nature of the primary literature on this topic, this data set provide valuable insight into the susceptibility of coral reef communities at intermediate depths. The experimental design and data analyses are sound, and the manuscript is well written. The fundamentals of this manuscript warrant publication, however, minor revisions are warranted in the opinion of this reviewer. Detailed comments have been included in the pdf using the comments feature.

The context of this study has been overstated and overgeneralized. The data for this study was collected at two similar depths (27 & 31m) boarding the upper mesophotic. To no fault of the authors, the upper depth limit of term “mesophotic” as coined by NOAA is poorly defined to correspond to the lower depth limits of recreational Scuba diving at 30-40m. This upper depth limit has no ecologically meaningful relevance to corals but pertains to a human (or socio-economic) limitation. The use of this data to generalize the susceptibility of mesophotic reefs (which can extend to depths >3X the depth of this study) to storm damage is a bit of a stretch. Understandably, while the authors attempted to collected data from a larger range of depths, the absence of data from shallower depths weakens their ability to generalize conclusions comparing the storm impact on “shallow” versus deep coral reefs.

While this study is valuable and detailed, previous studies have recorded storm induces damage to coral reefs at this depth and deeper (reviewed by Harmelin-Vivien 1994). The abstract of this manuscript implies that most literature to date has incorrectly characterized the susceptibility of deeper coral reefs to storm damage. The authors should reference the specific papers which have mischaracterized this issue to substantiate this implied generalization. The seminal reference on this topic (also referenced by the authors) clearly reviews these prior studies.

from Harlmelin-Vivien 1994
On several Pacific reefs, particularly in Guam and Hawaii, shallow outer reef areas (0-10 m) suffer less than deeper areas (15-25 m), as corals growing there are well adjusted in size and shape to strong hydrodynamic conditions, and endure better an increase in wave energy (RANDALL and ELDREDGE, 1977; 0GG and KosLow, 1978; DoLLAR, 1982; PFEFFER and TRIBBLE, 1985). Cyclone-induced damage generally decreases with depth but bt coral destruction was observed down to a depth of 25 m in Belize (HIGHSMITH et al., 1980), to 30m on the Great Barrier Reef (WoESIK et al., 1991), Guam (RANDALL and ELDREDGE, 1977) and Hawaii (WALSH, 1983), to 50 m in Jamaica (WooDLEY et al., 1981) and Hawaii (PFEFFER and TRIBBLE, 1985), and to 90-100 m in French Polynesia (LABOUTE, 1985; HAR¬ MELIN-VIVIEN and LABOUTE, 1986).

Experimental design

no comments

Validity of the findings

no comments

---

## Round 0.2 · accepted · Accept

Thank you for clearly addressing the suggestions of the referees in your revisions. I am happy to say that I believe your manuscript is a valuable contribution to the field, and can now be accepted for publication. Keep up the good work.